ecology, microbiology, plant science

priority effects, mutualism, symbiosis, *Medicago lupulina*, rhizobia, historical contingency

**Author for correspondence:**
Julia A. Boyle
e-mail: julia.boyle@mail.utoronto.ca

†These authors contributed equally to this work.

# Priority effects alter interaction outcomes in a legume–rhizobium mutualism

Julia A. Boyle[1], Anna K. Simonsen[2,3], Megan E. Frederickson[1,†] and John R. Stinchcombe[1,4,†]

[1]Department of Ecology and Evolutionary Biology, University of Toronto, 25 Willcocks Street, Toronto, Ontario, Canada M5S3B2
[2]Research School of Biology, Australian National University, Canberra, Australian Capital Territory 2601, Australia
[3]Department of Biological Sciences, Florida International University, Miami, FL 33199, USA
[4]Koffler Scientific Reserve, University of Toronto, Toronto, Ontario, Canada M5S3B2

JAB, 0000-0003-1029-8341; AKS, 0000-0002-5091-261X; MEF, 0000-0002-9058-7137; JRS, 0000-0003-3349-2964

Priority effects occur when the order of species arrival affects the final community structure. Mutualists often interact with multiple partners in different orders, but if or how priority effects alter interaction outcomes is an open question. In the field, we paired the legume *Medicago lupulina* with two nodulating strains of *Ensifer* bacteria that vary in nitrogen-fixing ability. We inoculated plants with strains in different orders and measured interaction outcomes. The first strain to arrive primarily determined plant performance and final relative abundances of rhizobia on roots. Plants that received effective microbes first and ineffective microbes second grew larger than plants inoculated with the same microbes in the opposite order. Our results show that mutualism outcomes can be influenced not just by partner identity, but by the interaction order. Furthermore, hosts receiving high-quality mutualists early can better tolerate low-quality symbionts later, indicating that priority effects may help explain the persistence of ineffective symbionts.

## 1. Introduction

Historical contingency often plays an outsized role in ecological communities, such that the order in which species arrive affects which species establish there [1]. Current and future species composition depends on what species were already there beforehand, with previous species being facilitative or inhibitive to new ones [1]. The effect of helping or harming future colonization is known as a priority effect, and these effects can manifest via abiotic (e.g. resource availability, space, etc.) or biotic pathways (e.g. competition, trophic levels, etc.) through destabilizing or equalizing mechanisms, such as niche modification or preemption [1,2]. Priority effects are often studied in the assembly of plant communities during succession or with invasive species [3]. There is less research on how priority effects influence microbial community assembly on a host, or how they change host–microbe interaction outcomes [4]. Priority effects are especially likely to occur if there is variation in a microbe's quality as a partner to their host. We tested if priority effects change the outcome of a plant–bacteria symbiosis by manipulating the order of arrival of different bacteria to a plant host, and then measuring the consequences for subsequent bacterial colonization and plant performance.

A classic assumption in microbial ecology is that bacteria are not dispersal-limited over space and time, described as the Baas-Becking hypothesis [5,6], and thus historical contingency is unimportant (see more in [7]). While some species of bacteria have global distributions, at a local scale the distribution of microbes can be patchy [8,9]. Newer work suggests species sorting, bacterial traits and relatedness, dispersal limitation, regional differences, and seasonal variation can all contribute to variation in bacterial communities in both water

and soil [10,11]. Variation in the bacterial communities of animal hosts is gaining interest; for example, *Caenorhabditis elegans* shows complex community assembly of their gut microbiome [12], and even the human gut microbiome is strongly shaped by diet and parental care when young, in ways suggestive of priority effects [13]. Just as animals host their own communities, plant hosts act as selective microcosms of their larger microbial landscape [4] and undergo complex community assembly of their own. Plant microbiomes such as the phyllosphere show evidence of priority effects when created synthetically [14]; however, the impact of priority effects on host performance and fitness remain unknown. In nature, there exist spatial differences in bacterial species [15] that can contribute to temporal differences in plant microbial communities [16], ultimately creating an environment where priority effects could occur.

Of the existing research on host-associated microbial priority effects, most centre on fungi–plant interactions [17]. Arbuscular mycorrhizal fungi show evidence of priority effects with the legume *Medicago truncatula*; when *M. truncatula* roots are already colonized, subsequent colonization decreases, and the decrease is more drastic, the longer the time gap is between fungus treatments [18]. However, Werner & Kiers [18] did not observe strong priority effects on plant performance from treatments applying two mutualistic fungi in different orders. Ectomycorrhizal fungi may also experience priority effects in colonizing their hosts when they are introduced with a time lag [19,20]; nonetheless, the effect on host performance remains less clear. These studies demonstrate the importance of priority effects to fungal microbial communities and raise the same question of whether priority effects are seen in other symbiotic microbes, such as nitrogen-fixing rhizobial bacteria and their legume hosts.

Legumes house rhizobacteria within specialized root structures called nodules, providing the bacteria with fixed carbon in exchange for fixed nitrogen. Most rhizobia are effective nitrogen-fixers, but not all are equally beneficial to their host plants [21], and there is a wide range of potential partners for the symbiosis. Effective nitrogen-fixing bacteria are mutualistic, while others may provide few benefits to hosts or overexploit plant resources [22,23]; however, less effective rhizobia can still be better than no rhizobia for hosts. The maintenance of variation in mutualist partners has long been identified as an important evolutionary question [24–26]. Coexistence theory describes how priority effects can determine the outcome of competition [27,28], and hence, the potential for low-quality partners to be maintained [29,30]. Despite the theoretical importance of priority effects, how and whether hosts mediate the outcome of priority effects between mutualistic symbionts is still poorly understood and requires empirical data. To our knowledge, no study has explicitly tested whether priority effects occur in one of the most well-studied and ecologically and economically important mutualisms, the legume–rhizobium symbiosis.

Here, we tested if the order of introduction of different nodule-forming bacteria strains (with known nitrogen-fixing qualities) affects bacterial colonization of hosts and host performance in the field. In the absence of priority effects, microbial communities on hosts should converge on the same abundance and identity of species, no matter which strain came first. Alternatively, effective nitrogen-fixing bacteria could facilitate plant colonization by other bacteria through niche modification, because plants with effective

rhizobia are more vigorous, have greater resources, and can thus support a higher biomass of later-colonizing bacteria, even if these secondary arrivals are non-beneficial or exploitative. Or, effective nitrogen-fixing symbionts might reduce the need for a plant to make more nodules, thus suppressing future nodulation and inhibiting colonization by other bacteria. The latter hypothesis is suggested by extensive research into the auto-regulation of nodulation [31–34], a signalling mechanism whereby legumes reduce nodule formation when they already have sufficient nitrogen. Ineffective nitrogen-fixing bacteria could also facilitate or inhibit future colonization by exploiting the plant, another instance of potential niche modification, as plants might accept more bacteria to fill their increased resource needs, or alternatively, ineffective symbionts might make the plant too small or weak to support other bacteria, or preemptively fill the root niche. Given these contrasting possibilities, the direction of any priority effects is an open empirical question.

## 2. Methods

### (a) Study system

We studied *Medicago lupulina*, an annual legume that forms indeterminate root nodules with rhizobia. Seeds were collected from the Koffler Scientific Reserve (KSR) in Ontario, Canada, in 2008, and to avoid maternal and plant genotype effects, in this experiment we used seeds from a single plant genotype that had been selfed for two generations in the University of Toronto greenhouses. We inoculated plants with two strains of bacteria with different nitrogen-fixing abilities: mutualistic *Ensifer meliloti* strain 1022 [35] and ineffective *Ensifer* sp. strain T173 [36]. *Ensifer meliloti* strain 1022 was first isolated from *Medicago orbicularis* growing wild in Greece [35], but the species has been found associating with *M. lupulina* in natural populations at KSR [37], while *Ensifer* T173 was first isolated in *Melilotus alba* growing wild in Canada, co-occurring in areas with *M. lupulina* [36]. As described in previous experimental work, *Ensifer* sp. strain T173 has a symbiotic plasmid, but poor symbiotic effectiveness, meaning that the many small, white nodules it forms on roots do not fix nitrogen [36] even when plants are fertilized with nitrogen or co-inoculated with other mutualistic strains [38]. Compared with uninoculated plants, plants inoculated with only *Ensifer* T173 showed increased mortality, reduced plant biomass, and increased time to flowering, demonstrated in both glasshouse conditions [38] and field conditions in previous inoculation experiments [39].

### (b) Seed preparation and planting

First, we scarified 270 *M. lupulina* seeds, immersed them in ethanol for 30 s, bleached them for 4 min, and then rinsed them in distilled water for 5 min. We next soaked the seeds in distilled water for another 30 min, before placing them on sterile agar plates which were left in a completely dark environment at 4°C. After one week of stratification, the seeds sat at room temperature for 12 h to germinate roots at least 1 cm long, and then we exposed them to sunlight for an hour to promote chlorophyll production. On 23 May 2019, we planted seeds individually into $10 \times 10 \times 10$ cm pots that had been bleach-sterilized, bottom-lined with landscape fabric to cover drainage holes and filled with autoclaved sand. Sand filled the pot to within 1.25 cm from the top, leaving a lip. We fertilized germinants twice: a 1 ml low-nitrogen fertilizer dose (recipe in the electronic supplementary material) on the same day they were planted in the greenhouse, followed by a second dose 16 days later. Pots were initially covered in

saran-wrap and kept in clear plastic bins in the KSR greenhouse; we used the saran-wrap and plastic bins to ensure high humidity during seedling establishment and reduce early colonization by non-focal microbes. On 7 June (15 days after germination), we planted the pots into holes in the ground in a natural grassy field at the KSR. We transplanted pots into a randomized, blocked design, with two spatial blocks of 135 pots each placed into a grid, with grid rows and columns separated by 30 cm. We placed pots into the holes so that the lip of the pot was 1.25 cm above ground, ensuring that the pot sand and exterior soil were flush. The lip provided a small barrier between the sand and any soil and water contamination.

## (c) Microbial treatments

Each bacterial inoculation contained the same cell densities (optical density, OD600, of 0.08 optical density units for all inoculations, a concentration of approx. $10^6$ cells ml$^{-1}$) of bacteria suspended in tryptone yeast (TY) liquid media. We gave each plant 1 ml of inoculum each round. To experimentally create the potential for priority effects, we separated the first and second rounds of inoculation by two weeks. We implemented a full $3 \times 3$ factorial design, with early inoculations of either *E. meliloti* 1022, *Ensifer* sp. T173, or a control TY media inoculation, fully crossed with later inoculations of either 1022, T173, or control TY media. Consequently, there were nine treatments with 30 plants per treatment. We added the first inoculation on 6 June 2019, the night prior to the plants being moved outside, and we then kept each of the bacterial treatment groups in separate bins overnight to avoid contamination among treatments. We administered the second inoculation on 20 June into the sand of the pots in the field, during a period of sunny days to avoid rain washing out the treatments. Bacteria in the soil may have colonized our experimental plants, but any significant treatment effects would suggest that experimental inoculations of bacteria nonetheless infected host plants, despite competition with soil bacteria.

## (d) Data collection

We counted leaves and checked for mortality after we placed pots in the field to monitor growth on 7, 17, 20 and 25 June and 7 and 22 July. The final leaf count was assessed by tracking the total number of leaves the plant produced throughout its life. We harvested plants on 22 July and kept them in a refrigerator until we collected nodules the following week. We harvested plants prior to flowering to avoid nodule senescence [40,41]. We counted all nodules from each plant that survived to the end of the experiment and identified them as containing effective or ineffective nitrogen-fixing bacteria based on colour, pink being indicative of nitrogen fixation and white indicating strain T173 (following [38]). Studying strain T173 in mixed inoculations, Simonsen & Stinchcombe [38] also scored nodule colour and then confirmed strain identities using antibiotic resistance assays, and the two methods of identifying which strain occupied a nodule were highly correlated ($r = 0.86$). They also found that co-infection of nodules was extremely infrequent (0.003% of cultured isolates), further suggesting that nodule colour is an accurate method of determining nodule occupancy of T173. Finally, we separated the aboveground biomass, dried it at 55°C for 48 h, and weighed it as an indicator of overall plant performance.

## (e) Data analysis

We analysed data in R v. 3.5.3 [42]. We archived code and data on Dryad [43] and include further information on the models in the electronic supplementary material. First, to evaluate if the order of inoculation affected plant mortality, we used two chi-squared tests with either first or second inoculation treatment as a predictor variable and the number of dead and alive plants as a response variable. For subsequent analyses, in general, we used linear models with the initial microbial treatment, the subsequent microbial treatment, and their interaction as fixed predictors; exceptions or additional covariates are described below. If the first strain or the interaction was significant, this suggested priority effects occurred. We initially included block as a random effect in linear models using the *lme4* [44] and *lmerTest* [45] packages, but subsequently excluded block when it explained zero variance. All ANOVAs were type III ANOVAs, calculated in the *car* package [46], unless otherwise specified. Plants that died during the experiment often had no aboveground biomass or nodules (excluded from relevant analyses); however, if roots were intact by the harvest date, the number of nodules could still be assessed. To improve normality in our nodule number model, we log-transformed nodule numbers, after adding 1 to make all values positive, non-zero numbers. To test the relative abundances of strains in nodules at the end of the experiment, we fitted a generalized linear model to the number of effective nodules versus the total number of nodules with a quasi-binomial distribution, weighted by the total nodule number. We also tried analysing these data as a MANOVA with pink and white nodules as potentially correlated response variables, and the results were qualitatively unchanged (electronic supplementary material, table S1). Additionally, we used *emmeans* [47] for an *a priori* planned contrast between plants receiving *E. meliloti* strain 1022 first followed by *Ensifer* sp. strain T173 and plants receiving these same strains in the opposite order. Comparing the 1022 first, T173 second treatment to the T173 first, 1022 second treatment tests for an effect of the order of arrival of strains (i.e. a priority effect) on the number of effective and ineffective nodules. Aboveground biomass had non-normal residuals even after log-transformation, so we used a generalized linear mixed model with a gamma error distribution instead.

## 3. Results

Mortality was significantly predicted by the first inoculation ($\chi_2^2 = 7.28$, $p < 0.05$), but the second inoculation had no effect ($\chi_2^2 = 0.121$, $p = 0.94$). Differences in mortality among treatments were mainly because of reduced mortality of plants treated with strain 1022 at the first inoculation. Plants that received 1022 as the first inoculum had 27.7% mortality, while those that received T173 first had 36% mortality, and those that received the control first had 46.6% mortality. The T173–1022 treatment had 40% mortality, compared with only 16.6% mortality in the 1022–T173 treatment (electronic supplementary material, table S2).

The total number of nodules on a plant differed depending on which strain it received first and the interaction between the first and second strains (table 1; electronic supplementary material, tables S3 and S4). Plants made significantly more nodules when given a bacterial inoculation compared with blank media, even when plants received the ineffective strain (figure 1). The uninoculated control plants had significantly fewer nodules than all other treatments (figure 1). However, a planned contrast found no significant difference in the number of nodules produced by plants in the 1022–T173 versus T173–1022 treatments ($p = 0.194$). Plants receiving T173 made more nodules than plants that did not receive T173, regardless of strain order (planned contrast, $p = 0.001$).

The number of effective and ineffective nodules differed based on the order of arrival, even between plants that got 1022 first and T173 second (83.0% effective nodules)

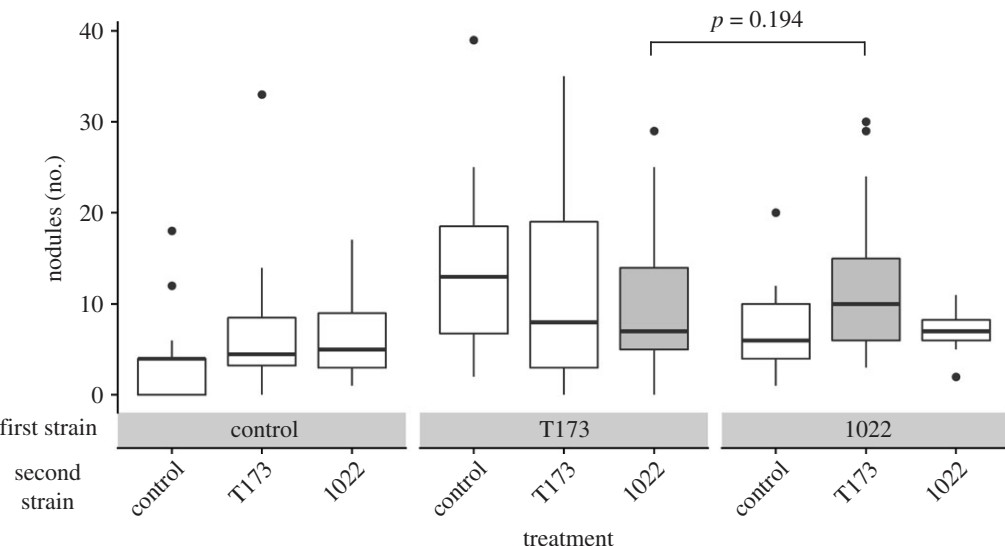

**Figure 1.** Number of nodules formed by plants inoculated with mutualistic *Ensifer meliloti* strain 1022 (1022), ineffective strain *Ensifer* sp. T173 (T173) or a sham inoculation with no bacteria (control) at the first and second time points. The boxplots show the median with the lower and upper hinges corresponding to the 25th and 75th percentiles, and the upper and lower whiskers represent the largest or smallest value, respectively, that is no further than 1.5 times the inter-quartile range from the hinge. The solid line compares the T173–1022 and 1022–T173 treatments (filled with grey) using a planned comparison ($p = 0.194$).

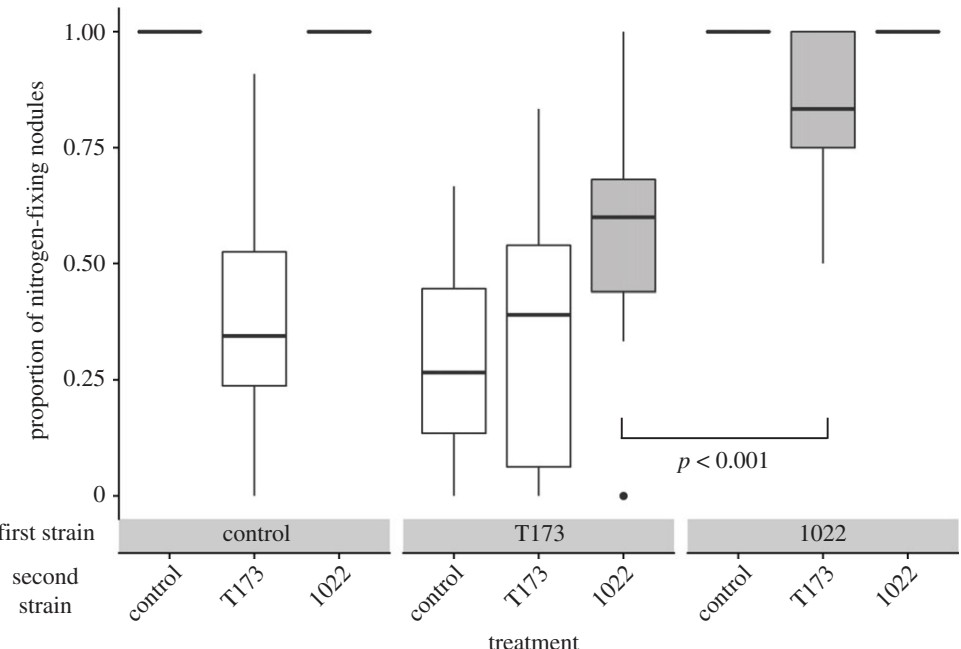

**Figure 2.** Proportion of nitrogen-fixing (i.e. pink) nodules formed by plants inoculated with mutualistic *Ensifer meliloti* strain 1022 (1022), ineffective strain *Ensifer* sp. T173 (T173) or a sham inoculation with no bacteria (control) at the first and second time points. The boxplots show the median with the lower and upper hinges corresponding to the 25th and 75th percentiles and the upper and lower whiskers represent the largest or smallest value, respectively, that is no further than 1.5 times the inter-quartile range from the hinge. The proportion was calculated by dividing the number of effective nodules by the total number of nodules for each plant. Only plants inoculated with T173 made any ineffective (i.e. white) nodules. The solid line compares the T173–1022 and 1022–T173 treatments (filled with grey) using a planned comparison of the proportion of effective nodules ($p = 0.0002$).

**Table 1.** Linear model results for nodulation and generalized linear mixed model (GLMM) results for plant performance. ($F$, $p$ and $\chi^2$ values are from type III ANOVAs. The GLMM used a gamma error distribution. Bold indicates statistically significant effects (i.e. $p \leq 0.05$).)

| predictors | total nodules (no.) | | | aboveground biomass (g) | | |
|---|---|---|---|---|---|---|
| | $F$ | d.f. | $p$ | Wald $\chi^2$ value | d.f. | $p$ |
| intercept | 1233 | 1, 178 | **<0.001** | 349 | 1 | **<0.001** |
| first strain | 10.5 | 2, 178 | **<0.001** | 16.4 | 2 | **<0.001** |
| second strain | 1.02 | 2, 178 | 0.364 | 0.422 | 2 | 0.810 |
| first strain × second strain interaction | 3.06 | 4, 178 | **0.0181** | 1.78 | 4 | 0.776 |

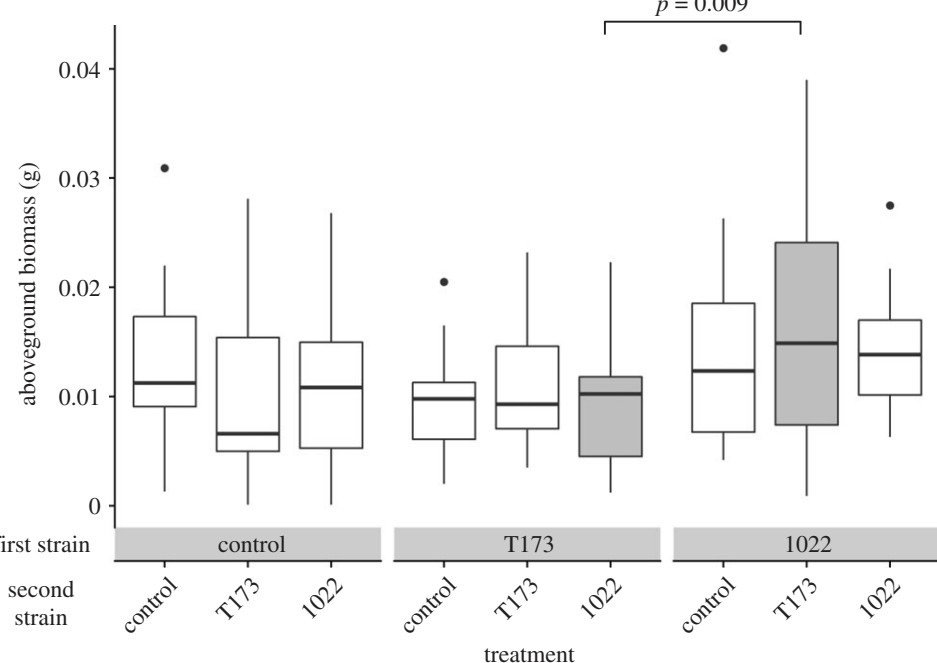

**Figure 3.** Aboveground biomass of plants inoculated with mutualistic *Ensifer meliloti* strain 1022 (1022), ineffective strain *Ensifer* sp. T173 (T173) or a sham inoculation with no bacteria (control) at the first and second time points. The boxplots show the median with the lower and upper hinges corresponding to the 25th and 75th percentiles and the upper and lower whiskers represent the largest or smallest value, respectively, that is no further than 1.5 times the inter-quartile range from the hinge. The solid line compares the T173–1022 and 1022–T173 treatments (filled with grey) in a planned comparison ($p = 0.009$). One outlier with a biomass over 0.05 g was omitted from this figure in treatment 1022–control.

**Table 2.** Generalized linear model (GLM) results for strain relative abundances. (Likelihood ratio $\chi^2$ and $p$ values are from the type III ANOVAs. The GLM used a quasi-binomial distribution. Bold indicates statistically significant effects (i.e. $p \leq 0.05$).)

| | strain relative abundances | | |
|---|---|---|---|
| predictors | likelihood ratio $\chi^2$ | d.f. | $p$ |
| first strain | 174 | 2, 183 | **<0.001** |
| second strain | 10.9 | 2, 183 | **<0.01** |
| first strain × second strain interaction | 30.9 | 4, 183 | **<0.001** |

and plants that got T173 first and 1022 second (57.6% effective nodules) (figure 2; electronic supplementary material, table S5). The proportion of nitrogen-fixing nodules was significantly predicted by the strain at time 1, strain at time 2 and their interaction (table 2; electronic supplementary material, table S6), even when accounting for the difference in the total nodule number. Therefore, the rhizobia strains a plant received, as well as the order in which they arrived, affected relative strain abundance. Notably, we found white nodules only on plants that received T173; this result further confirms that only T173 and not 1022 makes white nodules, because white nodules were never observed on plants that were not inoculated with T173. Plants that received only T173 had the highest proportion of ineffective (i.e. white) nodules (figure 2). Plants that received 1022 then T173 made significantly more effective nodules than plants that received T173 then 1022 (planned contrast, $p = 0.0002$);

thus, inoculation with 1022 first inhibited the subsequent formation of nodules with the ineffective strain, T173. When a plant was given a 1022 inoculation at any time, more than half of its nodules were likely to be effective, even if it received T173 earlier or later.

Aboveground biomass significantly differed depending on the first strain plants received (table 1; electronic supplementary material, table S7) with getting 1022 first leading to the highest aboveground biomass (figure 3; electronic supplementary material, table S8). The 1022–T173 treatment had the highest mean aboveground biomass, while the lowest was T173–1022, and this was significant in the planned contrast ($p = 0.009$). Receiving T173 first led to the lowest aboveground biomass, even if they were inoculated with 1022 later (figure 3; electronic supplementary material, figure S1). The number of nitrogen-fixing nodules was significantly, positively correlated with aboveground biomass (adjusted $R^2 = 0.277$, $p < 0.001$; electronic supplementary material, figure S2), but the number of ineffective nodules was not correlated with aboveground biomass (adjusted $R^2 = 0.0058$, $p = 0.15860$; electronic supplementary material, figure S3). As the proportion of effective nodules increased, so did aboveground biomass (electronic supplementary material, figure S4). Leaf number through time showed patterns similar to aboveground biomass and the rank order of treatments stayed consistent after inoculation with the first strain, suggesting little effect of the second strain (electronic supplementary material, figure S5).

## 4. Discussion

We found evidence of priority effects in rhizobial colonization of *M. lupulina* under field conditions, and these priority effects impacted the outcome of this plant–bacteria symbiosis. The

order of arrival of two *Ensifer* strains affected plant performance and the relative abundance of strains at the end of the experiment (i.e. final community composition), with the first strain added being the most influential. Furthermore, receiving the ineffective strain first made the mutualist strain less effective when it arrived later, while receiving the mutualist strain first prevented the ineffective strain from exploiting the plant and reducing plant performance.

Priority effects help or hinder future species colonization via two mechanisms: niche preemption or niche modification [1]. In our experiment, the two mechanisms are not easily distinguishable, and our results and logic suggest both could be occurring, where the niche being filled is the plant root and rhizobia are the colonizing species. We observed that the order of arrival of T173–1022 versus 1022–T173 did not affect how much of the niche was being used (i.e. the total number of infection sites on the roots), because the difference in total nodule number was non-significant between those treatments. Despite similar amounts of the niche being occupied, strain relative abundances differed, suggesting niche preemption, with one species preemptively using the resources that another species would need to successfully colonize [2,48,49]. However, niche modification may also have played a role, if the first strain to interact with the roots modified the root or host environment (e.g. through changes in nitrogen availability) in a way that changed subsequent nodulation. Regardless of the mechanism underlying the priority effect, our results are also congruent with the auto-regulation of nodulation (a process where legumes have an ability to reduce or limit the number of nodules they produce [50,51]), as both priority effects and autoregulation of nodulation suggest that past interactions between a host plant and microbes determine if later microbes can successfully colonize. When T173 was the first strain, it filled more of the available niche than when it arrived second, and vice versa for 1022 (figure 2). Nonetheless, the better strain (1022) was better at excluding T173 than the other way around, because the 1022–T173 had less than 20% ineffective nodules, while the T173–1022 had over 50% effective nodules (figure 2).

The implications of microbial priority effects for host benefits are important but have received little attention. Plant performance was determined by the first strain the plant received, suggesting that early exposure to symbionts is especially influential [52,53]. When plants have early exposure to effective nitrogen-fixers, plant performance is improved irrespective of the quality of later symbionts (figure 3), suggesting that priority effects help determine the benefits a microbiome confers to its host. Hosts, therefore, may be under strong selection to associate with an effective mutualist as soon as possible in their life cycle, potentially helping to explain why legumes secrete flavonoids to recruit compatible rhizobia [54]. Once associated, legumes with different quality symbionts may be considered as modified niches for microbes, with different nodulation responses (figure 1) and quantity and quality of resources [55]. Priority effects could, therefore, alter the local competitive dynamic of hosts with other plants, and hosts' interactions with other species; for example, increased plant nitrogen (provisioned by bacteria) may increase herbivory [39], or greater biomass may increase flower number and thus pollination [56]. On a broader scale, the composition and abundance of soil microbes are heterogeneous across space, and may change

seasonally [10], providing opportunities for priority effects to affect the success and distribution of plants across the landscape. When a plant colonizes a new site, associating with an effective partner first may mitigate costs of associating with less-effective partners later that may not be adapted to the host. By contrast, when a plant colonizes an environment where effective partners are less frequent, the plant could associate with ineffective partners first and experience priority effects that reduce its performance regardless of future effective partners, constraining plant distribution.

The result that microbial priority effects can influence the outcome of a mutualism is relevant to the maintenance of variation in mutualist quality. We know that high-quality partners may be maintained by positive fitness feedbacks or partner discrimination mechanisms [57,58], but the existence and maintenance of low-quality partners are less well understood [26]. Simonsen & Stinchcombe [39] previously showed that insect herbivores could help maintain low-quality partners, because herbivores preferentially attacked legumes inoculated with effective nitrogen-fixing rhizobia, thus eliminating fitness differences between plants with high- and low-quality bacterial partners. We have shown that priority effects at the level of an individual plant may promote larger-scale maintenance of variation in mutualist quality, because plant and microbial fitness depend not only on what microbes colonized each plant, but what order they arrived in. Ineffective symbionts extensively colonized *M. lupulina* roots, but only when they arrived early (figure 2). Furthermore, in our experiment, early exposure to an effective mutualist reduced the costs of acquiring an ineffective partner later (note similarity between 1022–1022 and 1022–T173 in figure 3), which may weaken selection against the ineffective partner and help it persist in the community. However, the opposite still occurs, where exposure to an ineffective partner first reduces the performance benefits of a good mutualist later on. Additionally, colonization–competition trade-offs [59] could support the idea that a poor-quality partner that makes many nodules but has low competitive ability in the soil, such as *Ensifer* sp. T173, can coexist alongside a higher-quality partner that makes fewer nodules but has higher competitive ability, such as *E. meliloti* 1022. A recent evolution experiment found that *E. meliloti* 1022 rapidly out-competed another low-quality strain [60], and non-nitrogen-fixing nodulating partners appear to be rare at our field site, given that some effective but no ineffective nodules were formed on control plants in our experiment (figure 2).

There were fairly low levels of colonization of rhizobia present in the field soil on our experimental plants, suggesting that our treatments were effective. Plants that received only sham inoculations had the fewest nodules, and all were pink, nitrogen-fixing nodules; this suggests that T173 did not travel between pots, and again, that the naturally occurring rhizobia in the soil are nitrogen-fixing. The absence of T173 in the environmental soil matches a previous study that sequenced naturally occurring rhizobia from *M. lupulina* nodules collected at the same field site [37] and found that most nodules contained *E. meliloti*, or another effective symbiont, *Ensifer medicae* [61]. Our results show that priority effects can occur under field conditions, and there are myriad ways that strains might arrive in different orders on plants in natural populations: for example, with rain moving rhizobia from site to site, existing patchiness of

soil microbes, or the death of a legume releasing a pulse of rhizobia from its roots. More research on time lags in the arrival of rhizobia on legumes in naturally occurring field communities, as well as how long priority effects persist in diverse legume species, is necessary to further develop our understanding of how priority effects function in plant–microbe symbioses. How priority effects play out in more diverse microbial communities in the soil, which often have more than just two rhizobial strains, remains to be seen, but our results suggest that plant fitness may be strongly shaped by which microbe(s) colonize it first.

Current intense interest in microbiomes means that it is more important than ever to understand how microbial communities assemble on hosts and how microbial community assembly affects interaction outcomes and host health. We showed that priority effects strongly influenced microbial colonization of hosts and the outcome of host–microbe interactions, with hosts benefiting more from getting an effective symbiont early. Most mutualisms and microbiomes are horizontally transmitted; they typically assemble anew on hosts that begin their lives with few, if any, microbes. Our results suggest that the first microbes to colonize hosts might have long-lasting effects, and largely determine host benefits and subsequent microbial community assembly.

Data accessibility. Data and code are available from the Dryad Digital Repository: https://doi.org/10.5061/dryad.8w9ghx3kz [43].

Authors' contributions. J.A.B., M.E.F. and J.R.S. collaborated on the ideas; J.A.B., M.E.F. and J.R.S. designed the methodology; J.A.B. performed the experiments; J.A.B., A.K.S., M.E.F. and J.R.S. wrote and edited the manuscript; and J.A.B., M.E.F. and J.R.S. analysed the data. All authors contributed to this manuscript and approve its publication.

Competing interests. We declare we have no competing interests.

Funding. This work was funded by the Natural Sciences and Engineering Research Council of Canada, specifically Discovery Grants to M.E.F. and J.R.S., and a USRA to J.A.B.

Acknowledgements. We thank Tia Harrison for her help and knowledge for all things *Medicago*, Dale Pebesma for assistance in the field, and the editor, one anonymous reviewer, and Gijsbert Werner for comments on the manuscript. We also thank Kate Brown, Will Sturch and John Jensen at the Koffler Scientific Reserve (2019), and Agriculture Canada, Jason Terpolilli and Rebecca Batstone for sharing rhizobial strains.

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
