## [Peer Review File · Proceedings of the Royal Society B: Biological Sciences]

Review History

RSPB-2020-2753.R0 (Original submission)

Review form: Reviewer 1

Recommendation

Major revision is needed (please make suggestions in comments)

Scientific importance: Is the manuscript an original and important contribution to its field?

Good

General interest: Is the paper of sufficient general interest?

Good

Quality of the paper: Is the overall quality of the paper suitable?

Good

Is the length of the paper justified?

Yes

Should the paper be seen by a specialist statistical reviewer?

No

Do you have any concerns about statistical analyses in this paper? If so, please specify them explicitly in your report.

Yes

It is a condition of publication that authors make their supporting data, code and materials available - either as supplementary material or hosted in an external repository. Please rate, if applicable, the supporting data on the following criteria.

Is it accessible?

Yes

Is it clear?

Yes

Is it adequate?

Yes

Do you have any ethical concerns with this paper?

No

Comments to the Author

Boyle et al. tackle an interesting question: How do priority effects occur in legume-rhizobia interactions in nature? Does it matter which rhizobia colonize roots first? This paper is well written and addresses an interesting and timely question. It is well-situated in the ecological and evolutionary literature. I appreciate the use of natural strains and the author's undertaking of a field experiment instead of a greenhouse or chamber experiment. I found the evidence that the first strain's identity drives plant biomass, nodule number, leaf number, and plant survival convincing. I also appreciated the organized code and data provided in the supplement for the statistical analysis.

While intriguing, I was less persuaded by the strain occupancy methodology and analysis. I believe the authors could do a better job articulating and supporting the assumption implicit in their study that nodule color is an accurate marker of strain identity and efficacy in mixed inoculations. Also, the data visualization and statistical methodologies employed could 1) better highlight the priority effects that are the focus of this manuscript and 2) tackle the challenging data distributions of the traits of interest. Lastly, using a single effective/ineffective strain contrast makes this study more appropriate as a proof of concept of priority effects rather than a study about the direction of influence of effective and ineffective strains on priority effects. Multiple pairs of effective and ineffective strains would need to have been used to make more general claims about directionality.

The strain identity assumption:

The authors make assumptions about strain identity based on nodule color in mixed inoculations. They assume that white nodules contain the ineffective strain T173, and pink nodules contain the effective strain 1022. This assumption needs to be better supported/discussed/addressed.

There are two obvious ways to provide support for the assumption that T173 inhabits white nodules. The first is to use sequencing of marker gene for a subset of the white and pink nodules or to conduct WGS shotgun sequencing on pools of pink and white nodules. Second, one could leverage the developmental trajectory of nodule formation during root development (e.g., the distribution of white nodules on roots). If white nodules contain T173, white nodules should be clustered at the bottom of roots in the 1022 -> T173 and control-> T173 treatment and at the top of roots in the T173 -> 1022 treatment.

While it is likely that T173 is the strain producing white nodules, there is an alternative hypothesis. The data showing that T173 produces white nodules come from single-strain studies. Nodule color and N-fixation are not always coupled. Strains that produce white nodules in an N-limited environment can produce pink nodules when fertilized with exogenous N. Mixed inoculations where some nodules host effective strains could be akin to the N-addition scenario. Thus, an alternative hypothesis for the shift from 83% pink nodules in 1022->T173 to 57% pink in T173->1022 is that when the effective strain is added first, the host forms nodules with the functional strain earlier thus creating an adequate N supply for leghemoglobin production in future nodules regardless of the identity of the strain that forms those nodules. Data showing that *M. lupulina* grown with T173 in N-rich conditions still produced white nodules would reduce this alternative hypothesis's likelihood.

Statistics:

It is not apparent why the reference level for the statistical models is the sham inoculation controls when the focus of the analysis centers on priority effects. Why not first check to see that the treatments have an effect above and beyond the controls, and then zoom in on the priority effects?

I am not sure the relative strain abundance 'models' summarized in Table 2 are the best way to test the author's relative strain occupancy hypothesis. As written, the authors conducted a MANOVA on effective and ineffective nodule numbers and separate linear models on log-transformed ineffective and effective nodule numbers. Such an analysis seems to me to be more focused on nodule numbers than strain relative abundance. It seems more parsimonious and statistically appropriate to do a single logistic regression (glm, family=binomial) that model successes (effective nodules) and failures (ineffective nodules). Such a model would more directly test for differences in the relative abundance of effective nodules while appropriately accounting for the large differences in nodule numbers (trials) between treatments. 6 of 12 nodules is very different statistically than 2 of 4 nodules even though both are at 50% relative abundance.

Figures:

The figures are not designed to emphasize the priority effects that are the focus of the abstract the framing of the paper. I am sure the authors tried many ways to clearly visualize their data (interactions are complicated!), but I'd proffer one additional suggestion. Consider reordering and renaming the levels on the x-axis this way (control->control | control->T173, T173->control, T173->T173, T173->1022 | 1022->control, control->1022, 1022->1022, 1022->T173). I found the multiple keys really difficult to follow.

Also, the transparency of figures could be improved by showing readers the data distributions. The symmetric standard errors currently used aren't appropriate for effective nodule percentages (when nodule numbers are changing across treatments) or the non-gaussian nodule number distributions (based on the supplemental figures). Individual data points or boxplots would be a far more effective data visualization tool. It will also create transparency in the replication differences caused by divergent survival between treatments.

Minor comments:

- The effect estimates for the ineffective nodule numbers analysis in Table 2 seem minuscule (e.g., 7.75×10^{-16}). If these effects are nearly zero, wouldn't it be clearer to round them and express them as 0.00?
- A Table, Figure, or Methods section describing the plant survival statistical model and analysis appears to be missing.
- The specific statistical models should be written out for readers in the supplement.
- Supplemental Figures 2 and 3 conflate the total nodule number on a plant with the number of effective nodules. It would be useful to see the relationship between (or lack thereof)

of the proportion of effective nodules on plants and biomass.

- I believe the Gore lab has done some work on priority effects in *C. elegans* guts that might be worth mentioning in the introduction or discussion.

Review form: Reviewer 2 (Gijsbert Werner)

Recommendation

Accept with minor revision (please list in comments)

Scientific importance: Is the manuscript an original and important contribution to its field?

Good

General interest: Is the paper of sufficient general interest?

Good

Quality of the paper: Is the overall quality of the paper suitable?

Good

Is the length of the paper justified?

Yes

Should the paper be seen by a specialist statistical reviewer?

No

Do you have any concerns about statistical analyses in this paper? If so, please specify them explicitly in your report.

No

It is a condition of publication that authors make their supporting data, code and materials available - either as supplementary material or hosted in an external repository. Please rate, if applicable, the supporting data on the following criteria.

Is it accessible?

Yes

Is it clear?

Yes

Is it adequate?

Yes

Do you have any ethical concerns with this paper?

No

Comments to the Author

This work studies priority effects among microbes in the plant-rhizobial mutualism. Ecologically, the roots potentially colonised by plant symbionts like rhizobia are a niche filled by those symbionts. As in other ecosystems, priority effects - or the order in which competing colonisers arrive at that niche - may influence the (relative) success of competing strains, and consequentially species community composition. In the context of a mutualistic interaction like here - this may also affect mutualism outcome, particularly when symbiont strains/species differ in their quality as a partner to the host.

Despite the ecological and evolutionary importance of plant root symbioses, priority effects have been very little studied in these system. There has been some work on plant-fungal mutualism, but very little work extending this to the other big plant-microbe mutualism, the plant-rhizobial. This study aims to begin filling that void, by using a factorial experimental design and two rhizobial strains differing in partner quality. The authors vary the order in which these strains are inoculated - representing early or later colonisation of the host - and measure nodules numbers (a proxy of rhizobial fitness), nodule fixation (a measure of partner quality/benefit to the host) and host biomass.

They find that the order in which rhizobia colonise a plant matters for nodule number, fixation and biomass - suggesting priority effects. Specifically they find that host benefits most when the high-quality partner colonises first. They suggest that this dynamic may contribute to explaining the continued existence of low-quality symbiont strains, despite the existence of partner discrimination mechanisms from plants.

This is an important study - to my knowledge the first to explicitly study and show priority effects in one of the world's most important mutualisms, the plant-rhizobial. While the existence of priority effects in this system may not in itself be surprising - for instance given that they have earlier been found in fungal plant root symbioses - the authors manage to quantify the size and direction of the effects - which was not obvious from prior theory or experimental work. They also manage to link the occurrence of priority effects with wider mutualistic outcomes

- and with the persistence of low-quality symbionts specifically - in an interesting and convincing way, bringing together two lines of research regarding mutualisms. The work also suggests various routes for future research - although the authors don't really make this explicit (adding suggestions on this in a new version of the manuscript could be relevant). A direction I could think of would be time period (magnitude of head start and/or persistence of priority effects) - because this was found to be a very important factor in the earlier work on priority effects in fungal symbionts.

I have only a few minor points for the authors to consider:

- lines 189-191: can the author make explicit what biological hypothesis they are testing with this a priori contrast - so the reader doesn't have to connect this back to the introduction/results.

- lines 252-267: the authors here discuss only niche preemption in detail, not niche modification. I don't think they want to suggest however, that niche preemption is the only causal mechanisms, nor do I think their design could really distinguish modification from preemption. If this is the case, adding a bit of discussion regarding niche modification may add a bit more balance to the paper? Or if they do want to argue for preemption (only) over modification, make that explicit and defend it accordingly.

- lines 273-274 ("Hosts may be...their lifecycle"). Can the authors give some suggestions at to how hosts could achieve this?

- I think the authors could/should add Carlstrom et al (2019) in Nature Ecology and Evolution (Synthetic microbiota reveal priority effects and keystone strains in the Arabidopsis phyllosphere) as potential reference for earlier work relevant to potential rhizobial priority effects.

Best wishes,
Dr. Gijssber Werner
University of Oxford
[\Signed]

Decision letter (RSPB-2020-2753.R0)

23-Dec-2020

Dear Miss Boyle:

Your manuscript has now been peer reviewed and the reviews have been assessed by an Associate Editor. The reviewers' comments (not including confidential comments to the Editor) and the comments from the Associate Editor are included at the end of this email for your reference. As you will see, the reviewers and the Editors have raised some concerns with your manuscript and we would like to invite you to revise your manuscript to address them.

Research ethics:

Use of animals and field studies:

It is a condition of publication that you make available the data and research materials supporting the results in the article. Please see our Data Sharing Policies (<https://royalsociety.org/journals/authors/author-guidelines/#data>). Datasets should be deposited in an appropriate publicly available repository and details of the associated accession number, link or DOI to the datasets must be included in the Data Accessibility section of the

article (<https://royalsociety.org/journals/ethics-policies/data-sharing-mining/>). Reference(s) to datasets should also be included in the reference list of the article with DOIs (where available).

Please submit a copy of your revised paper within three weeks. If we do not hear from you within this time your manuscript will be rejected. If you are unable to meet this deadline please let us know as soon as possible, as we may be able to grant a short extension.

Best wishes,
Dr Daniel Costa
mailto: proceedingsb@royalsociety.org

Associate Editor

Board Member: 1

Comments to Author:

Your manuscript has been assessed by two experts who find it good and potentially suitable for Proceedings B. However, one has valid concerns about data analysis and interpretation, and the other requests some improvements to presentation. Please note that Proceedings B does not allow multiple rounds of revision, so you must make every effort to address their concerns.

Reviewer(s)' Comments to Author:

Referee: 1

Comments to the Author(s)

Boyle et al. tackle an interesting question: How do priority effects occur in legume-rhizobia interactions in nature? Does it matter which rhizobia colonize roots first? This paper is well written and addresses an interesting and timely question. It is well-situated in the ecological and evolutionary literature. I appreciate the use of natural strains and the author's undertaking of a

field experiment instead of a greenhouse or chamber experiment. I found the evidence that the first strain's identity drives plant biomass, nodule number, leaf number, and plant survival convincing. I also appreciated the organized code and data provided in the supplement for the statistical analysis.

While intriguing, I was less persuaded by the strain occupancy methodology and analysis. I believe the authors could do a better job articulating and supporting the assumption implicit in their study that nodule color is an accurate marker of strain identity and efficacy in mixed inoculations. Also, the data visualization and statistical methodologies employed could 1) better highlight the priority effects that are the focus of this manuscript and 2) tackle the challenging data distributions of the traits of interest. Lastly, using a single effective/ineffective strain contrast makes this study more appropriate as a proof of concept of priority effects rather than a study about the direction of influence of effective and ineffective strains on priority effects. Multiple pairs of effective and ineffective strains would need to have been used to make more general claims about directionality.

The strain identity assumption:

The authors make assumptions about strain identity based on nodule color in mixed inoculations. They assume that white nodules contain the ineffective strain T173, and pink nodules contain the effective strain 1022. This assumption needs to be better supported/discussed/addressed.

There are two obvious ways to provide support for the assumption that T173 inhabits white nodules. The first is to use sequencing of marker gene for a subset of the white and pink nodules or to conduct WGS shotgun sequencing on pools of pink and white nodules. Second, one could leverage the developmental trajectory of nodule formation during root development (e.g., the distribution of white nodules on roots). If white nodules contain T173, white nodules should be clustered at the bottom of roots in the 1022 -> T173 and control-> T173 treatment and at the top of roots in the T173 -> 1022 treatment.

While it is likely that T173 is the strain producing white nodules, there is an alternative hypothesis. The data showing that T173 produces white nodules come from single-strain studies. Nodule color and N-fixation are not always coupled. Strains that produce white nodules in an N-limited environment can produce pink nodules when fertilized with exogenous N. Mixed inoculations where some nodules host effective strains could be akin to the N-addition scenario. Thus, an alternative hypothesis for the shift from 83% pink nodules in 1022->T173 to 57% pink in T173->1022 is that when the effective strain is added first, the host forms nodules with the functional strain earlier thus creating an adequate N supply for leghemoglobin production in future nodules regardless of the identity of the strain that forms those nodules. Data showing that *M. lupulina* grown with T173 in N-rich conditions still produced white nodules would reduce this alternative hypothesis's likelihood.

Statistics:

It is not apparent why the reference level for the statistical models is the sham inoculation controls when the focus of the analysis centers on priority effects. Why not first check to see that the treatments have an effect above and beyond the controls, and then zoom in on the priority effects?

I am not sure the relative strain abundance 'models' summarized in Table 2 are the best way to test the author's relative strain occupancy hypothesis. As written, the authors conducted a MANOVA on effective and ineffective nodule numbers and separate linear models on log-transformed ineffective and effective nodule numbers. Such an analysis seems to me to be more focused on nodule numbers than strain relative abundance. It seems more parsimonious and statistically appropriate to do a single logistic regression (glm, family=binomial) that model successes (effective nodules) and failures (ineffective nodules). Such a model would more directly

test for differences in the relative abundance of effective nodules while appropriately accounting for the large differences in nodule numbers (trials) between treatments. 6 of 12 nodules is very different statistically than 2 of 4 nodules even though both are at 50% relative abundance.

Figures:

The figures are not designed to emphasize the priority effects that are the focus of the abstract the framing of the paper. I am sure the authors tried many ways to clearly visualize their data (interactions are complicated!), but I'd proffer one additional suggestion. Consider reordering and renaming the levels on the x-axis this way (control->control | control->T173, T173->control, T173->T173, T173->1022 | 1022->control, control->1022, 1022->1022, 1022->T173). I found the multiple keys really difficult to follow.

Also, the transparency of figures could be improved by showing readers the data distributions. The symmetric standard errors currently used aren't appropriate for effective nodule percentages (when nodule numbers are changing across treatments) or the non-gaussian nodule number distributions (based on the supplemental figures). Individual data points or boxplots would be a far more effective data visualization tool. It will also create transparency in the replication differences caused by divergent survival between treatments.

Minor comments:

- The effect estimates for the ineffective nodule numbers analysis in Table 2 seem minuscule (e.g., 7.75 e-16). If these effects are nearly zero, wouldn't it be clearer to round them and express them as 0.00?
- A Table, Figure, or Methods section describing the plant survival statistical model and analysis appears to be missing.
- The specific statistical models should be written out for readers in the supplement.
- Supplemental Figures 2 and 3 conflate the total nodule number on a plant with the number of effective nodules. It would be useful to see the relationship between (or lack thereof) of the proportion of effective nodules on plants and biomass.
- I believe the Gore lab has done some work on priority effects in *C. elegans* guts that might be worth mentioning in the introduction or discussion.

Referee: 2

Comments to the Author(s)

This work studies priority effects among microbes in the plant-rhizobial mutualism. Ecologically, the roots potentially colonised by plant symbionts like rhizobia are a niche filled by those symbionts. As in other ecosystems, priority effects - or the order in which competing colonisers arrive at that niche - may influence the (relative) success of competing strains, and consequentially species community composition. In the context of a mutualistic interaction like here - this may also affect mutualism outcome, particularly when symbiont strains/species differ in their quality as a partner to the host.

Despite the ecological and evolutionary importance of plant root symbioses, priority effects have been very little studied in these system. There has been some work on plant-fungal mutualism, but very little work extending this to the other big plant-microbe mutualism, the plant-rhizobial. This study aims to begin filling that void, by using a factorial experimental design and two rhizobial strains differing in partner quality. The authors vary the order in which these strains are inoculated - representing early or later colonisation of the host - and measure nodules numbers (a proxy of rhizobial fitness), nodule fixation (a measure of partner quality/benefit to the host) and host biomass.

They find that the order in which rhizobia colonise a plant matters for nodule number, fixation and biomass - suggesting priority effects. Specifically they find that host benefits most when the

high-quality partner colonises first. They suggest that this dynamic may contribute to explaining the continued existence of low-quality symbiont strains, despite the existence of partner discrimination mechanisms from plants.

This is an important study - to my knowledge the first to explicitly study and show priority effects in one of the world's most important mutualisms, the plant-rhizobial. While the existence of priority effects in this system may not in itself be surprising - for instance given that they have earlier been found in fungal plant root symbioses - the authors manage to quantify the size and direction of the effects - which was not obvious from prior theory or experimental work. They also manage to link the occurrence of priority effects with wider mutualistic outcomes - and with the persistence of low-quality symbionts specifically - in an interesting and convincing way, bringing together two lines of research regarding mutualisms. The work also suggests various routes for future research - although the authors don't really make this explicit (adding suggestions on this in a new version of the manuscript could be relevant). A direction I could think of would be time period (magnitude of head start and/or persistence of priority effects) - because this was found to be a very important factor in the earlier work on priority effects in fungal symbionts.

I have only a few minor points for the authors to consider:

- lines 189-191: can the author make explicit what biological hypothesis they are testing with this a priori contrast - so the reader doesn't have to connect this back to the introduction/results.

- lines 252-267: the authors here discuss only niche preemption in detail, not niche modification. I don't think they want to suggest however, that niche preemption is the only causal mechanisms, nor do I think their design could really distinguish modification from preemption. If this is the case, adding a bit of discussion regarding niche modification may add a bit more balance to the paper? Or if they do want to argue for preemption (only) over modification, make that explicit and defend it accordingly.

- lines 273-274 ("Hosts may be...their lifecycle"). Can the authors give some suggestions as to how hosts could achieve this?

- I think the authors could/should add Carlstrom et al (2019) in Nature Ecology and Evolution (Synthetic microbiota reveal priority effects and keystone strains in the Arabidopsis phyllosphere) as potential reference for earlier work relevant to potential rhizobial priority effects.

Best wishes,
Dr. Gijssber Werner
University of Oxford
[\Signed]

Author's Response to Decision Letter for (RSPB-2020-2753.R0)

See Appendix A.

RSPB-2020-2753.R1 (Revision)

Review form: Reviewer 1

Recommendation

Accept with minor revision (please list in comments)

Scientific importance: Is the manuscript an original and important contribution to its field?

Excellent

General interest: Is the paper of sufficient general interest?

Good

Quality of the paper: Is the overall quality of the paper suitable?

Excellent

Is the length of the paper justified?

Yes

Should the paper be seen by a specialist statistical reviewer?

No

Do you have any concerns about statistical analyses in this paper? If so, please specify them explicitly in your report.

No

It is a condition of publication that authors make their supporting data, code and materials available - either as supplementary material or hosted in an external repository. Please rate, if applicable, the supporting data on the following criteria.

Is it accessible?

N/A

Is it clear?

N/A

Is it adequate?

N/A

Do you have any ethical concerns with this paper?

No

Comments to the Author

The authors have thoroughly and thoughtfully addressed reviewer comments. They have improved the statistical methods, the visualizations, and the justification of the assumption of strain identity. I believe this manuscript provides a useful contribution to the scientific community and a thoughtful and suitable contribution to Proc. B.

Note: I could not access the new Dryad repository to assess suitability.

Minor comments:

It would be helpful for reproducibility to interpret the Ensifer OD readings in terms of bacterial density.

The first paragraph of the discussion may have a typo: the data shows that the order of inoculation influences performance and occupancy, but it is not clear the order influences nodulation.

Decision letter (RSPB-2020-2753.R1)

10-Feb-2021

Dear Miss Boyle

I am pleased to inform you that your Review manuscript RSPB-2020-2753.R1 entitled "Priority effects alter interaction outcomes in a legume-rhizobium mutualism" has been accepted for publication in Proceedings B.

The referee(s) do not recommend any further changes. Therefore, please proof-read your manuscript carefully and upload your final files for publication. Because the schedule for publication is very tight, it is a condition of publication that you submit the revised version of your manuscript within 7 days. If you do not think you will be able to meet this date please let me know immediately.

To upload your manuscript, log into <http://mc.manuscriptcentral.com/prsb> and enter your Author Centre, where you will find your manuscript title listed under "Manuscripts with Decisions." Under "Actions," click on "Create a Revision." Your manuscript number has been appended to denote a revision.

You will be unable to make your revisions on the originally submitted version of the manuscript. Instead, upload a new version through your Author Centre.

- 1) A text file of the manuscript (doc, txt, rtf or tex), including the references, tables (including captions) and figure captions. Please remove any tracked changes from the text before submission. PDF files are not an accepted format for the "Main Document".
- 2) A separate electronic file of each figure (tiff, EPS or print-quality PDF preferred). The format should be produced directly from original creation package, or original software format. Please note that PowerPoint files are not accepted.
- 3) Electronic supplementary material: this should be contained in a separate file from the main text and the file name should contain the author's name and journal name, e.g. `authorname_procb_ESM_figures.pdf`

All supplementary materials accompanying an accepted article will be treated as in their final form. They will be published alongside the paper on the journal website and posted on the online figshare repository. Files on figshare will be made available approximately one week before the accompanying article so that the supplementary material can be attributed a unique DOI. Please see: <https://royalsociety.org/journals/authors/author-guidelines/>

4) Data-Sharing and data citation

It is a condition of publication that data supporting your paper are made available. Data should be made available either in the electronic supplementary material or through an appropriate

repository. Details of how to access data should be included in your paper. Please see <https://royalsociety.org/journals/ethics-policies/data-sharing-mining/> for more details.

<http://datadryad.org/submit?journalID=RSPB&manu=RSPB-2020-2753.R1> which will take you to your unique entry in the Dryad repository.

Once again, thank you for submitting your manuscript to Proceedings B and I look forward to receiving your final version. If you have any questions at all, please do not hesitate to get in touch.

Sincerely,
Dr Daniel Costa
Editor, Proceedings B
<mailto:proceedingsb@royalsociety.org>

Associate Editor Board Member: 1

Comments to Author:

There are three minor points raised by the reviewer that I think the authors can be trusted to address without further reviewer/editor checks.

Reviewer(s)' Comments to Author:

Referee: 1

Comments to the Author(s)

The authors have thoroughly and thoughtfully addressed reviewer comments. They have improved the statistical methods, the visualizations, and the justification of the assumption of strain identity. I believe this manuscript provides a useful contribution to the scientific community and a thoughtful and suitable contribution to Proc. B.

Note: I could not access the new Dryad repository to assess suitability.

Minor comments:

It would be helpful for reproducibility to interpret the Ensifer OD readings in terms of bacterial density.

The first paragraph of the discussion may have a typo: the data shows that the order of inoculation influences performance and occupancy, but it is not clear the order influences nodulation.

Decision letter (RSPB-2020-2753.R2)

11-Feb-2021

Dear Miss Boyle

I am pleased to inform you that your manuscript entitled "Priority effects alter interaction outcomes in a legume-rhizobium mutualism" has been accepted for publication in Proceedings B.

Your article has been estimated as being 9 pages long. Our Production Office will be able to confirm the exact length at proof stage.

Open Access

Paper charges

Sincerely,

Appendix A

Comments to Reviewers

We thank the editor and reviewers for their consideration and excellent feedback on our manuscript “Priority effects alter interaction outcomes in a legume-rhizobium mutualism”. We have incorporated their suggestions and improved our manuscript and its clarity. Specifically, we have addressed the concerns about strain occupancy, data analysis, and figure presentation. All our main results and conclusions remain intact after the revisions we made, including to statistical models. Detailed comments are included below, with author comments in bold and reviewer comments in plain text.

Associate Editor

Board Member: 1

Comments to Author:

Your manuscript has been assessed by two experts who find it good and potentially suitable for Proceedings B. However, one has valid concerns about data analysis and interpretation, and the other requests some improvements to presentation. Please note that Proceedings B does not allow multiple rounds of revision, so you must make every effort to address their concerns.

Thank you! We are pleased to hear it is potentially suitable for Proceedings B! We have endeavoured to be thorough in addressing the reviewers’ comments in this revision, and we provide detailed replies to their comments below.

Reviewer(s)' Comments to Author:

Referee: 1

Comments to the Author(s)

Boyle et al. tackle an interesting question: How do priority effects occur in legume-rhizobia interactions in nature? Does it matter which rhizobia colonize roots first? This paper is well written and addresses an interesting and timely question. It is well-situated in the ecological and evolutionary literature. I appreciate the use of natural strains and the author’s undertaking of a field experiment instead of a greenhouse or chamber experiment. I found the evidence that the first strain’s identity drives plant biomass, nodule number, leaf number, and plant survival convincing. I also appreciated the organized code and data provided in the supplement for the statistical analysis.

While intriguing, I was less persuaded by the strain occupancy methodology and analysis. I believe the authors could do a better job articulating and supporting the assumption implicit in their study that nodule color is an accurate marker of strain identity and efficacy in mixed inoculations. Also, the data visualization and statistical methodologies employed could 1) better highlight the priority effects that are the focus of this manuscript and 2) tackle the challenging data distributions of the traits of interest. Lastly, using a single effective/ineffective strain contrast makes this study more appropriate as a proof of concept of priority effects rather than a study about the direction of influence of effective and ineffective strains on priority effects.

Multiple pairs of effective and ineffective strains would need to have been used to make more general claims about directionality.

We thank the reviewer for the feedback! We have further justified using nodule colour to measure strain identity and responded to the helpful data visualization and modelling suggestions in detail below.

The strain identity assumption:

The authors make assumptions about strain identity based on nodule color in mixed inoculations. They assume that white nodules contain the ineffective strain T173, and pink nodules contain the effective strain 1022. This assumption needs to be better supported/discussed/addressed.

Excellent suggestion. We agree that this could have been more clear in the original manuscript. Several lines of evidence show that nodule colour is highly correlated with strain identity, even in mixed inoculations. We have now included a more thorough discussion of this evidence in the Methods (Lines 129-133, 179-184) and Results (Lines 234-236) of the paper. Briefly, a previous study that also used strain T173 in mixed inoculations (Simonsen and Stinchcombe, 2014 in Proc. R. Soc. B) found that strain identity inferred from nodule colour was highly correlated with strain identity inferred from culturing nodule bacteria on plates with and without antibiotics added (T173 is highly resistant to kanamycin and neomycin). We now describe this in the Methods (Lines 179-182). Furthermore, our own data strongly suggest that white nodules indicate the presence of T173 because we found zero white nodules on our experimental plants in the treatments that were not inoculated with T173 (this should now be more clear in the revised Figure 2, and at Lines 234-236 of the Results).

There are two obvious ways to provide support for the assumption that T173 inhabits white nodules. The first is to use sequencing of marker gene for a subset of the white and pink nodules or to conduct WGS shotgun sequencing on pools of pink and white nodules. Second, one could leverage the developmental trajectory of nodule formation during root development (e.g., the distribution of white nodules on roots). If white nodules contain T173, white nodules should be clustered at the bottom of roots in the 1022 -> T173 and control-> T173 treatment and at the top of roots in the T173 -> 1022 treatment.

We agree that sequencing would further confirm strain identities, but unfortunately we do not have sequence data at present and it would be very challenging to obtain them under the current circumstances. However, for reasons outlined below, our study and past experiments provide support and validation that nodule colour is a reliable indicator of strain occupancy.

While it is likely that T173 is the strain producing white nodules, there is an alternative hypothesis. The data showing that T173 produces white nodules come from single-strain

studies. Nodule color and N-fixation are not always coupled. Strains that produce white nodules in an N-limited environment can produce pink nodules when fertilized with exogenous N. Mixed inoculations where some nodules host effective strains could be akin to the N-addition scenario. Thus, an alternative hypothesis for the shift from 83% pink nodules in 1022->T173 to 57% pink in T173->1022 is that when the effective strain is added first, the host forms nodules with the functional strain earlier thus creating an adequate N supply for leghemoglobin production in future nodules regardless of the identity of the strain that forms those nodules. Data showing that *M. lupulina* grown with T173 in N-rich conditions still produced white nodules would reduce this alternative hypothesis's likelihood.

We have revised the paper to clarify that we know from previous work that T173 forms white nodules in both single- and mixed-inoculations, even under high nitrogen conditions. In the Bromfield et al. (2010) paper, they characterize single-inoculation T173 nodules as being small, numerous, ineffective, and white on *Medicago sativa*, *Melilotus alba*, *Medicago polymorpha*, and *Phaseolus vulgaris*. In the Simonsen & Stinchcombe (Proc. R. Soc. B, 2014) paper, T173 was added to *M. lupulina* both as a single strain and in a mixture with a mutualistic rhizobial strain. In the mixed inoculations by Simonsen and Stinchcombe, nodules were scored visually (based on the previous description by Bromfield et al.) and by antibiotic resistance assays; as we mentioned above and added to the paper, the two methods were highly correlated in determining nodule occupancy, and co-infection of nodules was extremely infrequent (this information has been included at Lines 181-184). Furthermore, when plants were supplemented with high-nitrogen fertilizer, T173 reduced host performance (compared to uninoculated controls) and still maintained the appearance of small, white nodules (Simonsen & Stinchcombe 2014) (described in Lines 129-133). This suggests that our visual scoring of T173 nodules was an accurate measure of nodule occupancy and that this strain is consistent in appearance across many different conditions.

We have now included the above information in our methods section (Lines 129-133, 179-184), and think it improves and better justifies our methods. We thank the reviewer for their thoughtful approach to this issue and suggestions.

Statistics:

It is not apparent why the reference level for the statistical models is the sham inoculation controls when the focus of the analysis centers on priority effects. Why not first check to see that the treatments have an effect above and beyond the controls, and then zoom in on the priority effects?

We take the reviewer's point, but when we consider other alternatives for reference level, they lead to more confusion. For example, if we choose strain 1022 as the reference group, then all treatments are compared to plants that got 1022 at both time points, when the interesting comparison in terms of priority effects is between plants that got 1022 first and T173 second and plants that got T173 first and 1022 second. Furthermore, we

chose the control (i.e., sham inoculation) treatment as the reference group at both time points because it is intuitive to compare everything to the uninoculated controls.

The reference level primarily affected our estimates in the models, and not the type III ANOVA outputs that we present in text. Because the estimates can be confusing based on which treatments are used as the reference point (especially for the interaction terms), and we and the reviewer had different reactions to what would be most appropriate, we have elected to remove them and present only model summary statistics in the main text (e.g., F statistics, df, p-values). We instead now provide predicted/least-squares, standard errors, standard deviations and sample sizes for all response variables for the 9 treatments in the supplementary material.

Finally, we also appreciate the reviewer's suggestion to first model the difference between uninoculated and inoculated plants, and then second model the difference between plants that got the two strain in different orders, but our experiment had a full-factorial design and if we fit two models to the dataset, it is not clear what to do with treatments that got sham inoculum at one time point and microbes at another (e.g., the Control-1022, 1022-Control, Control-T173, and T173-Control treatments). All this to say that we stuck with the full factorial 3 x 3 ANOVA with the double control (i.e., sham inoculum at both time points) as the reference group. Nonetheless, we think the revised presentation of the results in the main text and new supplementary tables (Tables S4, S6, S7) makes the model results much more clear.

I am not sure the relative strain abundance 'models' summarized in Table 2 are the best way to test the author's relative strain occupancy hypothesis. As written, the authors conducted a MANOVA on effective and ineffective nodule numbers and separate linear models on log-transformed ineffective and effective nodule numbers. Such an analysis seems to me to be more focused on nodule numbers than strain relative abundance. It seems more parsimonious and statistically appropriate to do a single logistic regression (glm, family=binomial) that model successes (effective nodules) and failures (ineffective nodules). Such a model would more directly test for differences in the relative abundance of effective nodules while appropriately accounting for the large differences in nodule numbers (trials) between treatments. 6 of 12 nodules is very different statistically than 2 of 4 nodules even though both are at 50% relative abundance.

We have now updated the way we statistically approached our nodule occupancy data. We replaced the MANOVA and two separate linear models with a single quasibinomial generalized linear model. We used the number of effective nodules with the total number of nodules as the response variable, and included a weights term for the total nodule number. This new model accounts for differences in total trials (e.g. total nodule number) that, as the reviewer states, is a statistical issue for our data.

The new model shows the same results as our MANOVA, so our main conclusions remain the same about nodule occupancy data in the text. We have included the

MANOVA results in the supplementary material (Table S1) as an alternative way of analysing the data, and focus on the new model in the main text. We thank the reviewer for their suggestions and believe this new statistical model provides a clear and appropriate way of analysing the data.

Figures:

The figures are not designed to emphasize the priority effects that are the focus of the abstract the framing of the paper. I am sure the authors tried many ways to clearly visualize their data (interactions are complicated!), but I'd proffer one additional suggestion. Consider reordering and renaming the levels on the x-axis this way (control->control | control->T173, T173->control, T173->T173, T173->1022 | 1022->control, control->1022, 1022->1022, 1022->T173). I found the multiple keys really difficult to follow.

We agree with the reviewer that the figures should be easier to interpret. We have modified the figures with the intent of emphasizing our priority effect treatments (1022-T173 and T173-1022); the boxplots for those treatments are now in a darker grey colour to stand out and the bar with significance values is easier to read visually. Instead of multiple keys, we have layered the x-axis with the main components of treatments (first strain and second strain). We visualized the order the reviewer suggested, but in the end, we decided to order the treatments primarily grouped by strain 1, since this had an important effect on the results of the experiment. We wanted to make the effect of the first strain very clear and easily interpretable in the graphs. We believe the changes are improvements to the overall accuracy and interpretability of the figures.

Also, the transparency of figures could be improved by showing readers the data distributions. The symmetric standard errors currently used aren't appropriate for effective nodule percentages (when nodule numbers are changing across treatments) or the non-gaussian nodule number distributions (based on the supplemental figures). Individual data points or boxplots would be a far more effective data visualization tool. It will also create transparency in the replication differences caused by divergent survival between treatments.

We thank the reviewer for their suggestion and have changed our main figures to be boxplots, which is indeed more in accordance with the data distributions.

Minor comments:

- The effect estimates for the ineffective nodule numbers analysis in Table 2 seem minuscule (e.g., 7.75×10^{-16}). If these effects are nearly zero, wouldn't it be clearer to round them and express them as 0.00?

These effect estimates have been removed from the tables, but all values are now clearly expressed.

- A Table, Figure, or Methods section describing the plant survival statistical model and analysis appears to be missing.

To make plant survival model descriptions more easily identifiable, we moved them to the beginning of the *Data analysis* section (Lines 188-190), and have provided a supplementary table (Table S2) with survival number and percent in each treatment.

- The specific statistical models should be written out for readers in the supplement.

We have now included a supplementary file with full and detailed written descriptions of each model, referenced in-text at Line 187. The R code will also be available on the Dryad repository.

- Supplemental Figures 2 and 3 conflate the total nodule number on a plant with the number of effective nodules. It would be useful to see the relationship between (or lack thereof) of the proportion of effective nodules on plants and biomass.

We have made an additional supplementary figure (Figure S4) with the proportion of nitrogen-fixing nodules on the x axis and aboveground biomass on the y-axis. We can see there is a positive linear relationship as the proportion increases. If we use a Gamma-distributed generalized linear model for aboveground biomass and as predictor use number of effective nodules versus total nodule number, with a weights term for total nodule number, then our 'proportion' effective nodules does significantly predict biomass ($p < 0.001$). This has been included in the R code on Dryad.

- I believe the Gore lab has done some work on priority effects in *C. elegans* guts that might be worth mentioning in the introduction or discussion.

We have added the relevant citation on *C. elegans* in the introduction to better contextualize the complex communities hosts harbour (Lines 66-67).

Referee: 2

Comments to the Author(s)

This work studies priority effects among microbes in the plant-rhizobial mutualism. Ecologically, the roots potentially colonised by plant symbionts like rhizobia are a niche filled by those symbionts. As in other ecosystems, priority effects - or the order in which competing colonisers arrive at that niche - may influence the (relative) success of competing strains, and consequentially species community composition. In the context of a mutualistic interaction like here - this may also affect mutualism outcome, particularly when symbiont strains/species differ in their quality as a partner to the host.

Despite the ecological and evolutionary importance of plant root symbioses, priority effects have been very little studied in these systems. There has been some work on plant-fungal mutualism,

but very little work extending this to the other big plant-microbe mutualism, the plant-rhizobial. This study aims to begin filling that void, by using a factorial experimental design and two rhizobial strains differing in partner quality. The authors vary the order in which these strains are inoculated - representing early or later colonisation of the host - and measure nodules numbers (a proxy of rhizobial fitness), nodule fixation (a measure of partner quality/benefit to the host) and host biomass.

They find that the order in which rhizobia colonise a plant matters for nodule number, fixation and biomass - suggesting priority effects. Specifically they find that the host benefits most when the high-quality partner colonises first. They suggest that this dynamic may contribute to explaining the continued existence of low-quality symbiont strains, despite the existence of partner discrimination mechanisms from plants.

This is an important study - to my knowledge the first to explicitly study and show priority effects in one of the world's most important mutualisms, the plant-rhizobial. While the existence of priority effects in this system may not in itself be surprising - for instance given that they have earlier been found in fungal plant root symbioses - the authors manage to quantify the size and direction of the effects - which was not obvious from prior theory or experimental work. They also manage to link the occurrence of priority effects with wider mutualistic outcomes - and with the persistence of low-quality symbionts specifically - in an interesting and convincing way, bringing together two lines of research regarding mutualisms. The work also suggests various routes for future research - although the authors don't really make this explicit (adding suggestions on this in a new version of the manuscript could be relevant). A direction I could think of would be time period (magnitude of head start and/or persistence of priority effects) - because this was found to be a very important factor in the earlier work on priority effects in fungal symbionts.

We thank the reviewer for their thoughtful feedback and suggestions. We have incorporated more on the future directions studies of priority effects may take (Lines 337-341).

I have only a few minor points for the authors to consider:

- lines 189-191: can the author make explicit what biological hypothesis they are testing with this a priori contrast - so the reader doesn't have to connect this back to the introduction/results.

We have made the biological hypothesis more explicit in this section of the methods (Lines 207-210) by specifying “Comparing the 1022 first, T173 second treatment to the T173 first, 1022 second treatment tests for an effect of the order of arrival of strains (i.e., a priority effect) on the number of effective and ineffective nodules.”

- lines 252-267: the authors here discuss only niche preemption in detail, not niche modification. I don't think they want to suggest however, that niche preemption is the only causal mechanisms, nor do I think their design could really distinguish modification from preemption. If this is the case, adding a bit of discussion regarding niche modification may add a bit more

balance to the paper? Or if they do want to argue for preemption (only) over modification, make that explicit and defend it accordingly.

We thank the reviewer for their suggestion and have updated the discussion to include more clarity on niche modification. In that paragraph we focused on nodule occupancy through the lens of niche preemption, and in the following paragraph we focus on plant response and performance through the lens of niche modification. Nodule occupancy is likely determined through a combination of both mechanisms, and as the reviewer states, indistinguishable in this study. We have added more on how nodule occupancy may be determined by both (Lines 265-267, 273-275).

- lines 273-274 ("Hosts may be...their lifecycle"). Can the authors give some suggestions as to how hosts could achieve this?

We have elaborated on how hosts may achieve this (Lines 288-290). Legumes secrete flavonoids to recruit compatible rhizobia, which may help them to associate with beneficial partners early on.

- I think the authors could/should add Carlstrom et al (2019) in Nature Ecology and Evolution (Synthetic microbiota reveal priority effects and keystone strains in the Arabidopsis phyllosphere) as potential reference for earlier work relevant to potential rhizobial priority effects.

We have added this paper to the introduction (Lines 71-73) and agree it improves the background of the manuscript, especially in the context of plant priority effects.

Best wishes,
Dr. Gijssber Werner
University of Oxford
[\Signed]